# Real-Time Tracking of Highly Luminescent Mesoporous Silica Particles Modified with Europium β-Diketone Chelates in Living Cells

**DOI:** 10.3390/nano11020343

**Published:** 2021-01-29

**Authors:** Jong-Seok Kim, Sung Ki Lee, Hansol Doh, Myeong Yun Kim, Do Kyung Kim

**Affiliations:** 1Myunggok Medical Research Institute, College of Medicine, Konyang University, Daejeon 35365, Korea; 2Department of Obstetrics and Gynecology, Konyang University Hospital, Daejeon 35365, Korea; sklee0728@konyang.ac.kr; 3Department of Food Science and Technology, University of California-Davis, Davis, CA 95616, USA; hdoh@ucdavis.edu; 4Department of Anatomy, College of Medicine, Konyang University Hospital, Daejeon 35365, Korea; my5960@nate.com

**Keywords:** mesoporous silica, europium complex, beta-diketone, live cell image, inflammation

## Abstract

Highly luminescent europium complexes modified mesoporous silica particles (MSP) were synthesized as an imaging probes for both in-vitro diagnostic and in-vivo cellular tracking agents. Europium β-diketone chelates (4,4,4-trifluoro-l-(2-thienyl)-l,3-butanedione) trioctylphosphine europium (III) (Eu(TTA)_3_(P(Oct)_3_)_3_) were incorporated inside the nanocavities that existed in hierarchical MSP (Eu@MSP). The MSP and Eu@MSP on mouse bone marrow-derived macrophages (BMDMs) did not show any toxic effect. The MSP and Eu@MSP in the BMDMs were found at cytoplasm without any degradation and immunogenicity. However, both pro- and anti-inflammatory cytokines of macrophages were significantly increased when lipopolysaccharide and a high concentration (100 μg/mL) of MSP and Eu@MSP were treated simultaneously.

## 1. Introduction

Nanoparticles have been widely used for nanomedicine applications [1]. Since it is possible to control the size and shape of the nanoparticles for their intrinsic and extrinsic characteristics such as an optical, electrical, and magnetic properties, the techniques for surface modifications and conjugation of biological ligands on nanoparticles have been increasing for bioimaging probes and theranostics both in in-vitro and in-vivo application [2,3]. Recently, mesoporous silica nanoparticles (MSN) have been noticed due to their physicochemical properties including consistent pore size, large surface area, and susceptibility of the surface functionality. These properties enable MSN to have a good biocompatibility and degradability [4]. In general, the mesoporous material has a tunable pore size in the range of 2–50 nm and it can regularly be oriented by structure. Also, the shapes and size of the MSN can be easily manipulated with the synthesis conditions such as a pH, type of surfactant including aromatic, aliphatic and heterocyclic, concentration of salts, and ration of solvents [5]. Therefore, MSN have been focused great attention in the field of pharmaceutical industry [6], boron neutron capture therapy (BNCT) [7], and fluorescence imaging probes [8].

Recently, versatile silica-based materials have been developed and it has been noticed with a great attention because researchers expect that these materials can overcome the issues which is about controlled release of hydrophobic drugs raised in biomedical field. According to the previous reports, many studies have investigated the use of the nanopore on–off system and this function is one of the unique and promising characteristics of MSN [9]. Silica is a well-known biocompatible material and has been used to enhance biocompatibility by altering the surface properties of implanted materials when exposed to the human body. Due to their high chemical loading capabilities, many attempts have been performed to release the therapeutic moieties in a controlled manner for prolonged drug delivery system (DDS) and mineralization potential [10]. Even though the silica-based particles have various advantages, they are hard to use as therapeutic molecules due to their high hydrophobicity. Instead of being modified with distinct formulation such as a form of capsule or conjugation with water soluble hydrophilic moieties, the mesoporous channels which are located inside the MSN can offer promising characteristics in perspective of improvement on physiochemical stability, controlled release, and prolonged drug bio-availability [11]. The advantages of MSN is that they have homogeneous particle size distribution and tunable pore size with TEOS monomer and surfactant, rather convenient to modify their surface with silane molecules.

Lanthanide (Ln^3+^)-based luminescent phosphorous particles are a strong candidate in the photovoltaic industry because of their intrinsic physiochemical properties such as high quantum yield with large Stoke shifts, anti-photobleaching, sharp emission spectrum, tunable emission band, high energy conversion efficient from UV region, and prolonged lifetime [12,13]. Due to their exceptional photosensitive and chemical properties, lanthanide-doped or lanthanide complexes modified particles could overcome the numerous difficulties related to organic based imaging dyes (references). Thus, these particles have been currently developed as exceedingly promising imaging probes for nanomedicinal applications. For instance, Ln^3+^-doped inorganic luminescent nanoparticles was developed as a donor for fluorescence resonance energy transfer (FRET) [14], bimodal (PET/MRI) imaging probe [15], and in-vitro analysis of tumor markers [16], MR contrast agents [17], cell-labeling and tracking [18], animal imaging [19], drug delivery system [20], photodynamic therapy (PDT) [21], and photothermal therapy [22]. Therefore, integration of lanthanide complexes into cavities of MSN has enhanced the photo- and thermal stabilities and against photobleaching [23]. Especially, Eu doped hydroxyapatite shows good antimicrobial activities [24,25,26].

According to the previous reports, lanthanide complexes can be integrated into mesoporous materials either by modest doping process or by covalent bond on it [27]. The nanocaged platforms with MSP are focused as a novel drug-carrier due to their intrinsic functions in which therapeutic molecules can be loaded as a cargo. After arrival of the drug-loaded MSP at a target legion by macrophage, the therapeutic molecules are released in a controlled manner by their characteristic gating phenomena (one-off switch) via osmatic and capillary pressure. Finally, MSP carrier vehicles are decomposed in a physiological condition. For this reason, the decoration of MSP by functional ligands has had significant consideration. In general, direct incorporation of europium complex into silica aerogel was attempted without any covalent bond [28,29].

Lately, nanoparticles have been implemented with a wide range of approaches aimed at therapeutically-enhancing or modifying the function of immune cells. Nanoparticles are used as drug carriers to improve delivery efficiency and specificity to immune cells [30]. In particular, macrophages are the most central cells that uptake nanoparticles. Because they control the polarization of macrophages, it is a critical point to examine the effect of newly-synthesized nanoparticles on macrophages [31].

In this work, MSP was used for incorporating the β-diketone europium complexes inside of their nanocavities. The MSP was prepared by emulsion template process and the procedure was conducted for tailoring its surface with functional ligands, resulting in strong chelation of Eu(TTA)_3_(P(Oct)_3_)_3_ by covalent bond. First, both surface and nanocavities of MSP were modified with a silane coupling agent, i.e., 3-(trimethoxysilyl)propyl methacrylate (TMSPMA) as an intermediary to activate of acryloyl group that is crucial for further functionalization of carboxylic acid with methacrylic acid by condensation reaction with potassium persulfate as an initiator. The cytotoxicity and uptake of MSP and Eu@MSP were examined against several mammalian cells, including macrophages, and investigated their effect on cytokine secretion, an essential function of macrophages.

## 2. Materials and Methods

### 2.1. Materials

Tetraethyl orthosilicate (TEOS), 3-(trimethoxysilyl)propyl methacrylate (TMSPMA), europium chloride hexahydrate, methacrylic acid (MAA), trioctylphosphine (P(Oct)3) were purchased from Sigma-Aldrich. Cetyl trimethylammonium bromide (CTAB), Potassium persulfate (PPS), sodium styrene sulfonate (SSS) ethanol (EtOH), ammonium hydroxide (NH_4_OH, purity) were purchased from Samchun chemicals (Kyunggido, South Korea). 4,4,4-trifluoro-1-(2-thienyl)-1,3-butanedione (TTA) was supplied by Tokyo Chemical Industry Co., Ltd. (Tokyo, Japan). All the chemicals were used without any further purification. Double distilled and deionized water was used throughout the study and it was prepared by ELGA Flex 3 Water Purification System Midland VWS Co., Ltd. (Cannock, UK).

### 2.2. Characterization

The morphological properties of the synthesized MSP were evaluated by JEM-ARM200F field emission transmission electron microscopy (FE-TEM, 200 kV, JEOL, Seoul, Korea) and JSM-7610F field emission scanning electron microscope (FE-SEM, 10 kV, JEOL, Seoul, Korea). Bruker ALPHA FT-IR (Billerica, MA, USA) spectrometer equipped with platinum ATR with diamond module was used for finding the chemical structures of nanoparticles. The scan parameters were 256 scans with a resolution of 1 cm^−1^ at the wavenumber range of 400~4000 cm^−1^. Excitation and emission spectra of the samples were examined with using a RF-5301PC (Shimadzu, Kyoto, Japan) photoluminescent equipped with a 150 W xenon lamp. DLS particle size and zeta potential measurements were performed on the monomodal and multimodal samples using a Malvern Zetasizer HSA 3000 (Malvern, UK). Thermal gravimetric analysis (TGA) and differential scanning calorimetry (DSC) were used for investigating thermodynamics of the nanoparticles (Shimadzu TGA-50 and DSC-50, Kyoto, Japan) in the range from 20 °C to 700 °C with a heating rate of 10 °C/min.

### 2.3. Preparation of Mesoporous SiO_2_ (mSiO_2_) Particles (MSP)

MSP was synthesized by condensation of hydrolysis of TEOS and CTAB [6]. Briefly, CTAB (1.56 g), EtOH (303 mL), ddH_2_O (132 mL), and NH_4_OH (25 g) were added in 1-L flask and sonicated for 10 min until fully dissolved. The solution was magnetic stirred for 30 min at room temperature (600 rpm). Then, TEOS (5 mL) was added and reacted at 70 °C for 2 h. The residues were obtained through centrifugation at 4000 rpm for 10 min. After centrifugation, the residue was fully dried and transferred to glass vial. Then the sample was calcinated to remove the organic compounds in furnace at 550 °C for 5 h and collect the MSP. The MSP was dispersed in ethanol (EtOH) with a concentration of 50 mg/mL. To activate the acryloyl group of MSP, 500 μL of TMSPMA was added under the condition of sonication for 10 min. Diluted acetic acid (1 mL, 10 wt% in water) was added and reacted for 10 min. The sample was collected by centrifuge at 4000 rpm for 10 min and decantated for obtaining the residues. The washing process was repeated for 3 times. After drying out the solvent, H_2_O (10 mL) was added for dispersing under the sonicated condition for 10 min. To activate the carboxylic acid, 20 μL of MMA, 100 mM of SSS (500 μL), and 130 mM of PPS (500 μL) were added and the temperature was increased up to 70 °C with magnetic stirring for 1 h. The excess reactants and impurities were removed by centrifugation for three times and the carboxylic acid activated MSP (MSP-COOH) was dispersed in EtOH for further chelation with europium complexes.

### 2.4. Synthesis of β-Diketone Eu(TTA)_3_(P(Oct)_3_)_3_ Complexes

β-diketone europium complex Eu(TTA)_3_(P(Oct)_3_)_3_ is prepared with slight modification of previous study [32]. The synthetic route is illustrated in Figure 1. Briefly, europium (20 mM), TTA, and P(Oct)_3_ stock solutions were dissolved in 50 mL of EtOH. Eu(TTA)_3_(P(Oct)_3_)_3_ complexes were synthesized with 12 mL of 20 mM TTA and 0.1 mL of 1 M NH_4_OH followed by addition of 12 mL of 20 mM P(Oct)_3_ and added the dropwise with 4 mL of 20 mM Eu^3+^ in 50 mL glass vial with a cap. The glass vial was located in water bath under magnetic stirring at 60 °C for 1 h.

### 2.5. Synthesis of Eu(TTA)_3_(P(Oct)_3_)_3_ Doped MSP (Eu@MSP)

Eu(TTA)_3_(P(Oct)_3_)_3_@MSP (Eu@MSP) were synthesized with the dropwise addition of Eu(TTA)_3_(P(Oct)_3_)_3_ (1 mL) into 50 mL of MSP with activating carboxyl group on the surface. When the addition of Eu(TTA)_3_(P(Oct)_3_)_3_ complexes to carboxyl group activated MSP, the temperature was kept at 60 °C for 30 min. The sample was dialyzed against three changed of 1 L distilled water using membrane tubing (MWCO = 12,500) and concentrated under reduced pressure. The final chemical structure of Eu@MSP is shown in Figure 1.

### 2.6. Cell Lines and Cell Culture

A549 and HeLa cell line were obtained from Korean Cell Line Bank (KCLB; Seoul, Korea). All cells were cultured in Dulbecco’s modified Eagle’s medium (DMEM; Nuaille, France) supplemented with 10% heat-inactivated fetal bovine serum (FBS, Biowest), 100 units/mL penicillin, and 100 μg/mL streptomycin at 37 °C in a humidified incubator containing 5% CO_2_.

### 2.7. Isolation of Bone Marrow Cells and Generation of Murine Bone Marrow-Derived Macrophage Generation

Animal Care and the Guiding Principles for Animal Experiment Using Animals were approved by the University of Konyang Animal Care and Use Committee (19-23-A-01). Bone marrow cells were obtained from the femur and tibia of 6–8 week-old female C57BL/6 mice (DBL, Chungbuk, Korea). Murine bone marrow-derived macrophages (BMDMs) were obtained as described in previous study [33]. Briefly, female C57BL/6J mice were euthanized by CO_2_ asphyxiation. Bone marrow cells were cultured for 6 days in DMEM (Biowest) containing 100 U/mL penicillin, 100 μg/mL streptomycin, 10% FBS (Biowest), and 20 ng/mL recombinant mouse macrophage colony stimulating factor (M-CSF, R&D Systems) at 37 °C in the presence of 5% CO_2_. Non-adherent cells were removed and differentiated macrophages were incubated in antibiotic-free DMEM until use.

### 2.8. CCK-8

The cytotoxicity was determined using a Cell Counting Kit-8 (Dojindo, Kumamoto, Japan). The cells were plated at 96-well plates (0.5~1 × 10^4^ cells/well) and incubated overnight. After treating with MSP and Eu@MSP for 24 h, the cells were washed with PBS and changed to a fresh medium with 10% CCK-8. Afterward, the samples were incubated at 37 °C for 2 h. The absorbance was measured at the wavelength of 450 nm using a microplate spectrophotometer (BioTek, Winooski, VT, USA).

### 2.9. Flow Cytometry Analysis

Annexin V Apoptosis Detection Kit I (BD Pharmingen, San Diego, CA, USA) was used for identifying the apoptotic and necrotic cells. The BMDM cells were seeded in a 6-well plate (6 × 10^5^ cells/well) for overnight, and then the cells were harvested after 24 h incubation with MSP and Eu@MSP. The nanoparticle treated cells were washed with ice-cooled PBS, and resuspended in 0.5 mL of Annexin V binding buffer. Then, the cells (1 × 10^5^ cells/well) were stained with 5 µg/mL of propidium iodide and 5 µL of Annexin V-FITC in 50 µL Annexin V binding buffer at 4 °C. After 10 min, 300 µL of binding buffer was added to the samples for investigating using a CytoFLEX (Beckman Coulter, Indianapolis, IN, USA) and FlowJo (Tree Star, Ashland, OR, USA).

### 2.10. Fluorescence and Holotomographic Microscopy

To find out the MSP and Eu@MSP were well internalized to cells, 10 μg/mL of MSP and Eu@MSP were incubated with BMDM. After 4 h incubation, cells were washed three time with PBS and fixed with 3.7% formaldehyde with PBS for 15 min at room temperature. Then, the sample was permeabilized in 0.05% of Triton X-100 and stained actin with Rhodamine Texas Red (Invitrogen, Carlsbad, CA, USA). After washing, the specimens were mounted in FluoroshiledTM with DAPI (Sigma, St. Louis, MO, USA). Eu@MSP, actin, and nucleic acid were observed using a Ts2-FL fluorescence microscope (Nikon, Kyoto, Japan). Holotomographic images were obtained using a 3D Cell Explorer-fluo (Nanolive, SA, Ecublens, Switzerland) microscope equipped with a 60 × magnification (λ  =  520 nm, sample exposure 0.2 mW/mm^2^) and a depth of field of 30 µm.

### 2.11. ELISA

Cell culture supernatant were collected and stored at −80 °C until use. The levels of TNF-α, IL-6, and IL-10 were determined by ELISA using a commercial reagent kit following the manufacturer’s instruction (eBioscience, San Diego, CA, USA).

## 3. Results

### 3.1. Preparation and Characterization of Eu@MSP

MSP is synthesized by hydrolysis of TEOS with rod-shape micelle forming liquid crystal as a template (cationic surfactant and CTAB in the presence of H_2_O/EtOH) (Figure 1). After removing the CTAB by calcination, acryloyl group was activated with TMSPMA in the presence of acetic acid. To activate the carboxylic acid, MMA, SSS, and PPS as a catalysis are added and temperature was increased to 70 °C. β-diketone europium complex, Eu(TTA)_3_(P(Oct)_3_)_3_ was prepared by mixing the stoichiometric compositions of Eu^3+^, TTA and P(Oct)_3_ in EtOH. Eu(TTA)_3_(P(Oct)_3_)_3_@MSP (Eu@MSP) were synthesized by dropwise addition of Eu (TTA)_3_(P(Oct)_3_)_3_ carboxylic acid activated MSP.

The morphological variation of the samples was investigated with SEM and TEM. TEM images of MSP and europium complexes conjugated mSiO_2_ (Eu@MSP) are presented in Figure 2. MSP shows the uniform spherical shape and poor mesopores with defect-free crystal structure. The diameter of MSP is slightly increased from 400 to 600 nm compared to the Eu@MSP. The gap between the MSP is clearly shown in Figure 2a due to the steric hinderance between the particles, while the neck between the Eu@MSP is appeared as shown in Figure 2e. The necking substance is obviously the organic coating molecules and the pores in MSP are efficiently packed with fluorescent β-diketone Eu^3+^ chelates. The surface of Eu@MSP is rather flat than that of MSP is quite rugged surface. Appendix A shows the normalized DLS particle size distribution curves for MSN is around 580 nm and that of TMSPMA@mSiO_2_, MMA@mSiO_2_, and Eu@MSP presents about 643 nm, 689 nm, and 705 nm of average diameter respectively. The MSP and Eu@MSP have negative zeta potentials of approximately −20.4 mV and −16.4 mV.

TGA analysis was performed to investigate the thermal decomposition of MSP, TMSPMA@MSP, MMA@MSP, and Eu@MSP. The MSP shows extremely stable thermal behavior with a very little weight loss of about 0.25% up to 700 °C due to the elimination of CTAB (Figure 3a). TMSPMA@MSP presents around 4% of weight loss of when temperature went up to 172 °C due to the decomposition of TMSPMA evaporation of the moisture, and residual reactants. As temperature increased up to 700 °C, the further weight loss is occurred about 1%, which could be attributed from the decomposition of TMSPMA molecules into carbon dioxide and silica layer. In general, MSP possess extremely high surface area of more than 1000 m^2^/g. Therefore, it can be inferred that the amount of TMSPMA molecules on MSP surface is less than 1% and it is almost attached as a monolayer on the surface of MSP. In the case of MMA@MSP, the weight loss (8.5%) was caused by the decomposition of MMA and TMSPMA when temperature went up to 250 °C. As temperature was increased to 700 °C, the further weight loss is around 2%. The inorganic/organic hybrid composites of Eu@MSP have two weight loss stages; the loss is about 14% until 200 °C can be attributed to evaporation of the water molecules, residual reactant moieties, and solvents; decomposition amount of organic compounds was approximately 3% between 200 and 500 °C is raised from the continuous degradation of soft segments, which is constituent molecules of Eu@MSP, i.e., Eu^3+^ ions are composed of metal complexes with tertiary phosphine, P(Oct)_3_, carbonyl group from TTA and carboxyl group from MMA covalently bonded on MSP.

Figure 3b shows the DSC thermograms of MSP, TMSPMA@MSP, MMA@MSP, and Eu@MSP. Based on the existence of crystal structures resulting from the step-wise modification process for Eu@MSP, and DSC analysis was done to investigate the glass transition temperature (*T_g_*) and crystallin melting point of the Eu@MSP. Four major thermal transition peaks were emerged for Eu@MSP; two endothermic thermal transition peaks at 65.7 and 211 °C were allocated to *T_g_* and the crystalline melting temperature (T_m_) of Eu@MSP, respectively. The exothermic peak at 307 °C can be assigned to thermal decomposition of P(Oct)_3_.

Figure 4 shows the result of the ATR-FTIR measurements for MSP, TMSPMA@MSP, MMA@MSP, Eu@MSP, and Eu(TTA)_3_(P(Oct)_3_)_3_. In the case of MSP, the peak at 1065 cm^−1^ corresponding to Si-O-Si asymmetric and the peak at 810 cm^−1^ corresponding to Si-O-Si symmetric vibrations. The peaks at 1410 and 1352 cm^−1^ are assigned to ν(C=C, C=S thienyl heterocycle). CF_3_ in TTA moiety vibration peaks are appeared at 1292 cm^−1^ (ν(CF_3_)) and 723 cm^−1^ (δ(CF_3_)). The vibration peaks of keto-enol tautomerization of β-diketone ligand can be assigned to 1535 cm^−1^ (ν(C=O)) and 1516 cm^−1^ (ν(C=C)). The peaks at 1064 cm^−1^ (δ(O-CH_3_)), 861 cm^−1^ (δ(CH_3_)), and 797 cm^−1^ (δ(CH, thienyl)). The new peak appeared at 1638 cm^−1^ in TMSPMA@MSP spectrum in Figure 4b is allocated to the axial deformation of the C=C termination of TMSPMA molecules.

### 3.2. Photoluminescent Properties of Eu (TTA)_3_(P(Oct)_3_)_3_@mSiO_2_

Figure 5a presents the photoluminescent (PL) spectra of Eu@MSP depended on the concentration. The PL spectra of Eu@MSP exhibits the distinctive characteristic emission peaks of f-f transition of Eu^3+^ complexes with a strong red emission. As shown in Figure 5a,b, Eu@MSP shows the maximum excitation wavelengths at 308 nm (λ_em_ = 625 nm) and the maximum emission wavelengths at 625 nm (λ_ex_ = 308 nm). The sharp excitation peaks appeared at 308 nm is coming from the absorption of spherical silica nanoparticles not from the energy transfer from europium complexes. In the case of emission (excitation (λ^em^ = 625 nm) spectrum for the ^5^D_0_
→
^7^F_j_ transition of Eu(TTA)_3_(P(Oct)_3_)_3_, four emission peaks ascribed to ^5^D_0_
→
^7^F_j_ (Eu^3+^, J = 0–4) transitions are monitored at 583, 597, 622, and 662 nm under excitation at 308 nm.

### 3.3. Effect of MSP and Eu@MSP on the Cytotoxicity of Various Kinds of Eukaryotic Cells

To evaluate the cytotoxic effects of MSP and Eu@MSP, human lung cancer cell A549, cervical cancer cellar HeLa, mouse bone marrow cell, and mouse bone marrow-derived macrophages (BMDMs) were selected. Incubation with MSP and Eu@MSP for 24 h in the range of the concentration between 1 μg/mL and 100 μg/mL did not show any toxic effects on the A549 and HeLa cells (Figure 6a,b). In addition, there were no cytotoxicity is shown by the MSP and Eu@MSP on mouse bone marrow cell (Figure 6c,d). These results are confirmed with flow cytometry and the result also showed that there were no cytotoxic effect of MSP and Eu@MSP (Figure 6e).

### 3.4. Cellular Uptake of MSP and Eu@MSP

The study of whether nanoparticles could make entry into cells is of great importance for the design of harmless and efficient nano-medicines. Therefore, a fluorescent microscope, flow cytometry, and holotomography are used to check whether MSP and Eu@MSP could enter the cells. There are no nanoparticles in the HeLa cells and A549 cells, but the uptake of Eu@MSP can be recognized in the BMDMs (Figure 7a). These results are further confirmed using flow cytometry by analyzing Forward versus Side scatter (FSC vs. SSC), which is commonly used to identify cells of interest based on size and internal complexity/granularity.

As shown in Appendix A, in the BMDMs control group, most of the cells are in the lower-left panel. In contrast, when treating nanoparticles, many cells are found in the upper panel, depending on the concentration. These results indicate that the nanoparticles uptake to BMDMs, which increases the specific internal complexity or granularity of the cells and is detected in a high panel in flow cytometry.

A live cell holotomography was used to classify the intracellular entry of MSP and Eu@MSP into the BMDMs. As macrophages are phagocytic cells which are known to engulf various forms of nanoparticles, the live cell holotomography data shows that the MSP and Eu@MSP could enter the macrophage by phagocytosis. In particular, the macrophages had active movements of pseudopodia, and the nanoparticles near the pseudopodia were recognized and entered into the cells by phagocytosis. The MSP and Eu@MSP in the BMDMs were found at the cytoplasm without degradation for several hours, and there was no difference between MSP and Eu@MSP (Figure 7b and Appendix A).

### 3.5. Effect of MSP and Eu@MSP on the Secretion of Cytokine by BMDMs

Previous studies demonstrated that nanoparticles could interact with various immune cells such as macrophage and dendritic cells and either enhance or inhibit function. To investigate whether MSP and Eu@MSP affect the cytokine secretion by BMDMs, MSP and Eu@MSP were treated with or without LPS to BMDMs for 24 h, and measured the amount of TNF-α, IL-6, and IL-10. LPS stimulation via Toll-like receptor 4 can lead to macrophage maturation and secretion of pro- and anti-inflammatory cytokines. As shown in Figure 8a–c, MSP and Eu@MSP treatments without LPS did not affect the secretion of cytokines by BMDMs. In addition, the effect of MSP and Eu@MSP particles on cytokine secretion of macrophages by LPS treatment was investigated. Co-treatment with LPS did not affect the any cytokine secretion at concentration up to 10 μg/mL. There is no serious effect on IL-6 secretion (Figure 8a–c). The text continues here.

## 4. Discussion

The MSP has a lot of benefits in perspective of having homogeneous and manageable pore size and properties that easy to modify its internal and external surface with varieties of silane derivatives. Europium β-diketone chelates (4,4,4-trifluoro-l-(2-thienyl)-l,3-butanedione) trioctylphosphine europium (III) Eu(TTA)_3_(P(Oct)_3_)_3_ were successively incorporated inside of the cavities in MSP. After removing the CTAB by calcination, the acryloyl group was activated with TMSPMA in the presence of acetic acid. The surface of MSP was activated with functional ligands and it led to strong chelation of Eu(TTA)_3_(P(Oct)_3_)_3_ by covalent bond. The final form of Eu@MSP had a fascinating core–shell structure, wherein the core possessed inorganic substance with mesopores and the shell was composed of β-diketone Eu^3+^ complexes. According to the result of SEM observation, the morphological properties of MSP which before and after coating with an organometallic compound were similar to those shown in the TEM observations. The particle size distribution of modified MSP showed the larger dimensions which means the coating moieties absorbing on the surface of MSN, and the dimension of the MSP was eventually increased. For biomedical applications, the surface charge of MSN should be in the proper ranges. The particles will be agglomerated if zeta potential is too low, while the particle will have a strong affinity with the cellular membrane if the zeta potential is too high. When the concentration of Eu(TTA)_3_(P(Oct)_3_)_3_ against MSP is increased, _5_D^0^
→
_7_F^2^ transition peak at 625 nm is linearly increased with an almost Gaussian shape. Figure 5c shows the excitation (λ_em_ = 625 nm) spectra for the ^5^D_0_
→
^7^F_j_ transition of Eu(TTA)_3_(P(Oct)_3_)_3_ and Eu@MSP. The excitation (λ_em_ = 625 nm) spectrum of Eu(TTA)_3_(P(Oct)_3_)_3_ reveals the broad excitation band between 250 and 450 nm with sharp maximum peaks at 308 nm and 350 nm, which can be allocated to the energy absorption of ligands by transferring energy via “ligand-to-metal” pattern [12]. However, the excitation (λem = 625 nm) spectrum of Eu@MSP shows the disappearance of the peak at 350 nm due to the interference of MSP. ^5^D_0_ → ^7^F_2_ transition peak is the main contribution of typical photoluminescence of Eu^3+^ complexes with a bright red emission color under UV light. Comparing to Eu(TTA)_3_(P(Oct)_3_)_3_, ^5^D_0_
→
^7^F_2_ transition peak of Eu@MSP is read shift from 622 nm to 625 nm and Eu^3+^ could be existed as a more polarized resulting in relatively remarkable hypersensitive phenomena. The peak position shift was caused by asymmetrical coordination and confined status of Eu^3+^ complexes influenced by surrounding MSP.

The live cell holotomography shows that the intracellular uptake of MSP and Eu@MSP into the BMDMs. TNF-α and IL-10 secretion of BMDMs are dramatically increased when lipopolysaccharide (LPS) is treated with MSP and Eu@MSP at high concentration (100 μg/mL) at the same time. Macrophages have dynamic movements of pseudopodia, and the nanoparticles near the pseudopodia are identified and uptake by phagocytes. The MSP and Eu@MSP in BMDMs are found at cytoplasm without degradation for several hours, and there is no difference in both MSP and Eu@MSP. These results demonstrate that MSP and Eu@MSP could enhance LPS-induced secretion of TNF-α and IL-10 by BMDMs. As a result, the MSP and Eu@MSP do not have any immunogenicity itself, but high concentrations of MSP and Eu@MSP can increase the secretion of pro-inflammatory and anti-inflammatory cytokines from macrophages when treated with LPS.

The Eu@MSP can be applicable if the surface of the particles is modified with bio-ligands, i.e., antibodies, aptamers, proteins, and carbohydrates, in the field of bioimaging probes both in-vivo and in-vitro tracking agents. Especially, we expect that Eu@MSP shield the antigens coming from enzymatic degradation together with fast denaturation, resulting in an extended discharge of antigen for long-term humoral response, or to target oriented controlled release for specific cell immunity. In addition, Eu@MSP could enhance the antigen uptake by antigen-presenting cells (APC), promoting a stronger immune response resulting in improving therapeutic effects.

## Figures and Tables

**Figure 1 nanomaterials-11-00343-f001:**
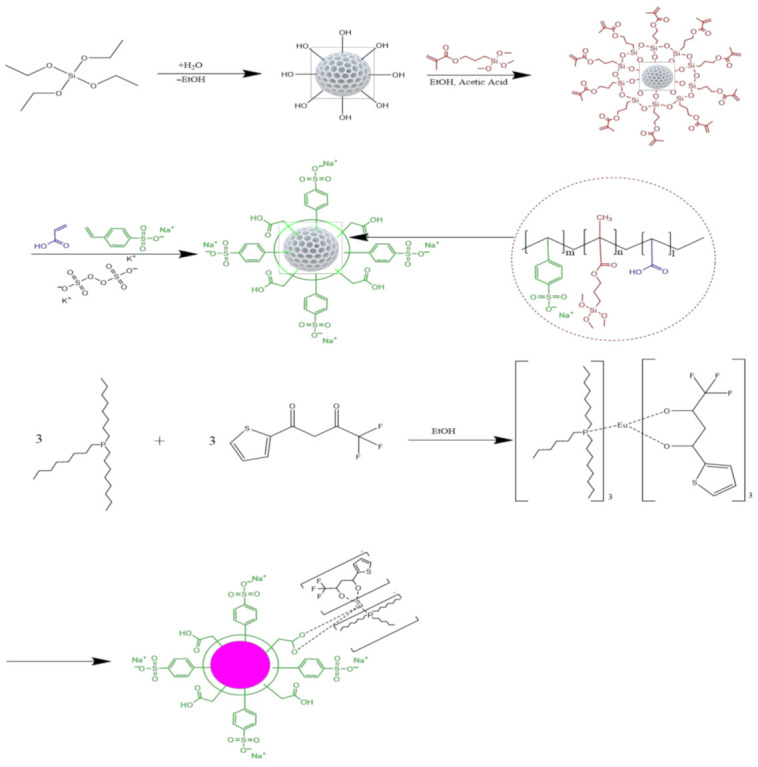
Schematic flow diagram of the synthesis route for Eu(TTA)_3_(P(Oct)_3_)_3_ modified MSP by multi-step process. First, both surface and nanocavities of mesoporous silica particles (MSP) were modified with a silane coupling agent, i.e., 3-(trimethoxysilyl)propyl methacrylate (TMSPMA) as an intermediary to activate of acryloyl group. To activate the carboxylic acid, MMA, sodium styrene sulfonate (SSS), and potassium persulfate (PPS) as a catalysis are added and temperature was increased to 70 °C. β-diketone europium complex, Eu(TTA)_3_(P(Oct)_3_)_3__,_ was prepared by mixing the stoichiometric compositions of Eu^3+^, TTA and P(Oct)_3_ in EtOH. Eu(TTA)_3_(P(Oct)_3_)_3_@MSP (Eu@MSP) were synthesized by dropwise addition of Eu(TTA)_3_(P(Oct)_3_)_3_ carboxylic acid activated MSP.

**Figure 2 nanomaterials-11-00343-f002:**
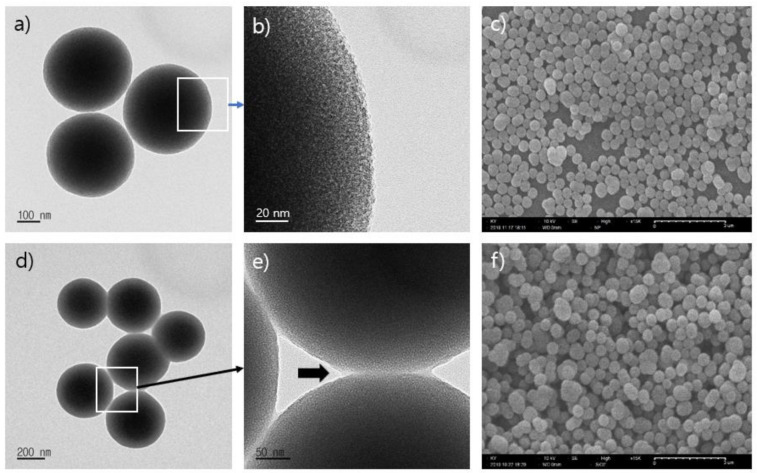
Transmission electron micrograph (TEM) images of (**a**) mesoporous silica particles (mSiO_2_); (**b**) enlarged image of rectangle region in a); (**c**) scanning electron micrograph (SEM) image of mSiO_2_ particles; TEM images of (**d**) europium complexes, Eu(TTA)_3_(P(Oct)_3_)_3_, decorated mSiO_2_ (Eu(TTA)_3_(P(Oct)_3_)_3_@mSiO_2_); (**e**) enlarged image of rectangle region in d); (**f**) SEM image of Eu(TTA)_3_(P(Oct)_3_)_3_@mSiO_2_.

**Figure 3 nanomaterials-11-00343-f003:**
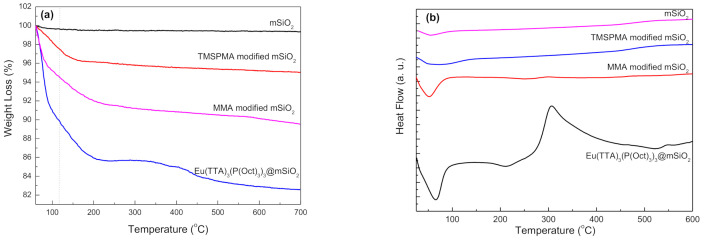
(**a**) Thermal gravimetric analysis (TGA) curves and (**b**) differential scanning calorimetry (DSC) thermograms of step-by-step modified samples for mSiO_2_, TMSPMA modified mSiO_2_ (TMSPMA@mSiO_2_), MMA modified mSiO_2_ (MMA@mSiO_2_) and Eu(TTA)_3_(P(Oct)_3_)_3_@mSiO_2_.

**Figure 4 nanomaterials-11-00343-f004:**
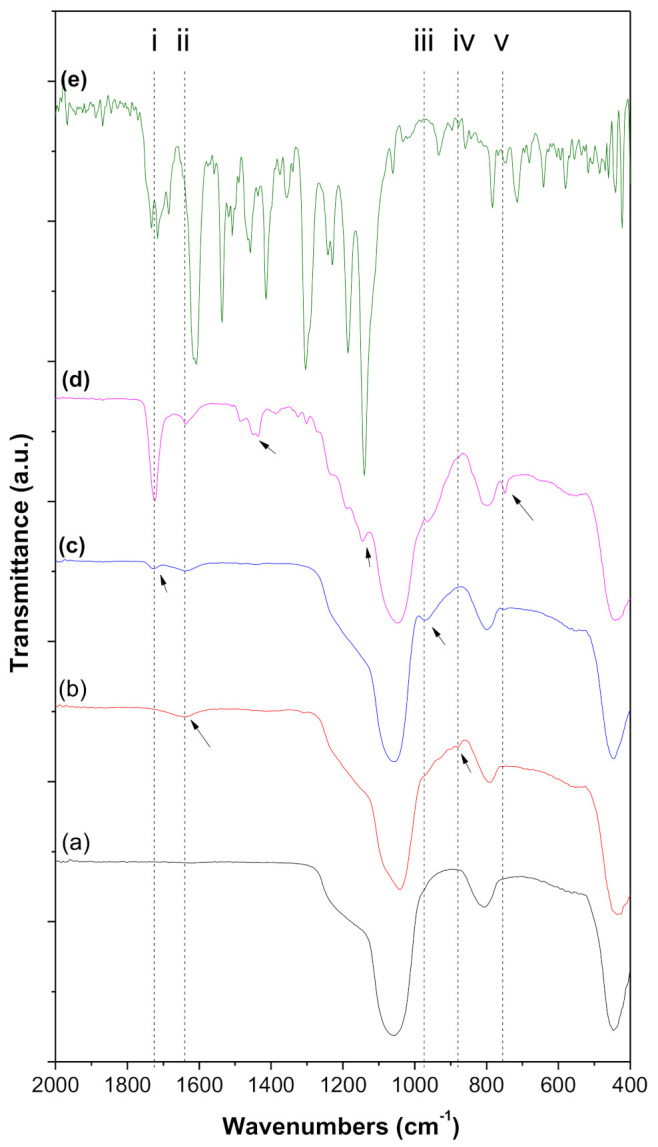
FTIR spectra of step-by-step modified samples for (**a**) mSiO_2_ calcined at 550 °C for 5 h; (**b**) TMSPMA modified mSiO_2_ (TMSPMA@mSiO_2_); (**c**) MMA modified mSiO_2_ (MMA@mSiO_2_); (**d**) Eu(TTA)_3_(P(Oct)_3_)_3_@mSiO_2_; and (**e**) Eu(TTA)_3_(P(Oct)_3_)_3_.

**Figure 5 nanomaterials-11-00343-f005:**
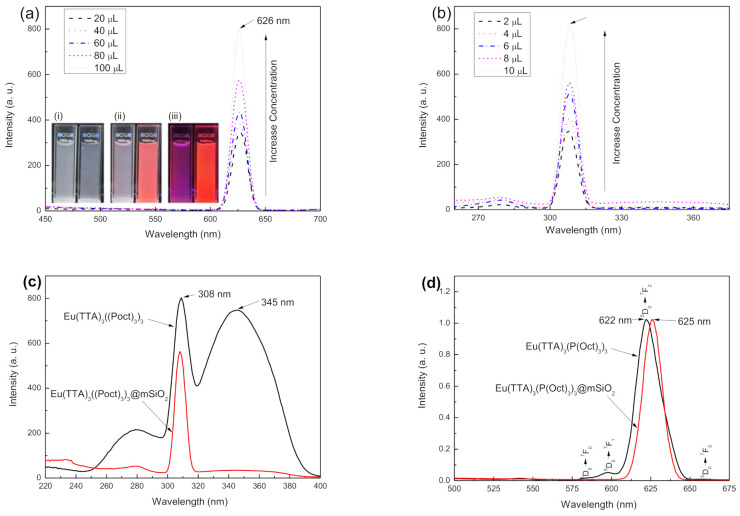
Concentration dependent (**a**) emission (λ_ex_ = 308 nm) and (**b**) excitation (λ_em_ = 625 nm) spectra for the ^5^D_0_
→
^7^F_j_ (Eu^3+^, J = 0–4) transition of Eu(TTA)_3_(P(Oct)_3_)_3_@mSiO_2_. (**c**) excitation (λ_em_ = 625 nm) spectra for the ^5^D_0_
→
^7^F_j_ transition of Eu(TTA)_3_(P(Oct)_3_)_3_ and Eu(TTA)_3_(P(Oct)_3_)_3_@mSiO_2_. (**d**) emission (λ_ex_ = 308 nm) spectra for the ^5^D_0_
→
^7^F_j_ (Eu^3+^, J = 0–4) transition of Eu(TTA)_3_(P(Oct)_3_)_3_ and Eu(TTA)_3_(P(Oct)_3_)_3_@mSiO_2_. Inserted images in (**a**) are mSiO_2_ (left) and Eu(TTA)_3_(P(Oct)_3_)_3_@mSiO_2_ (right) exposed under (i) room light, (ii) room + UV light and (iii) UV light.

**Figure 6 nanomaterials-11-00343-f006:**
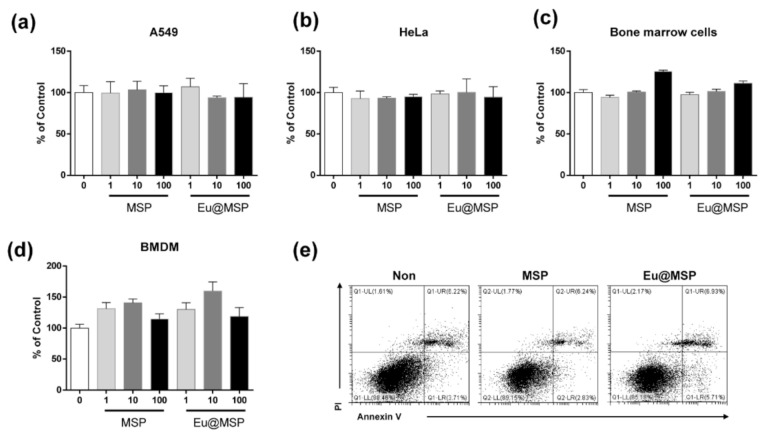
Cytotoxicity of MSP and Eu@MSP to A549, HeLa, mouse bone marrow cells and mouse bone marrow-derived macrophage. (**a**) A549 (**b**) HeLa (**c**) mouse bone marrow cells and (**d**) BMDMs were plated in 96-well culture plates (1 × 10^4^ cells/well). After 24 h, the cells were incubated with 1, 10, and 100 μg/mL of MSP and Eu@MSP, respectively, for 24 h. Cytotoxicity was evaluated using CCK-8 as described in the Materials and Methods section (n.s, no significant). (**e**) Detection of BMDMs cell death assessed by flow cytometry using FITC-Annexin V and PI double immunostaining. BMDMs were untreated, treated with MSP and Eu@MSP for 24 h.

**Figure 7 nanomaterials-11-00343-f007:**
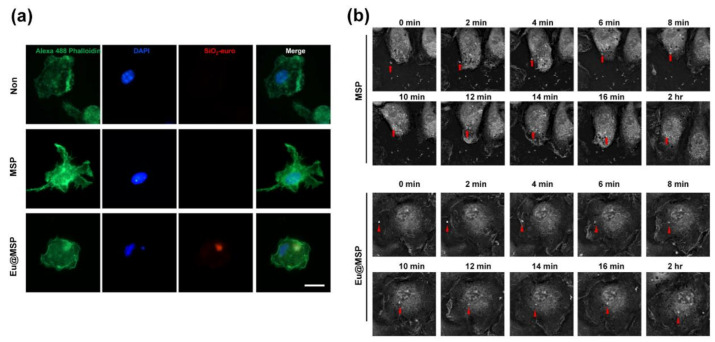
MSP and Eu@MSP uptake by mouse bone marrow-derived macrophages (BMDMs). (**a**) Intracellular staining of europium red-coupled mSiO_2_ in BMDMs (Scale bar = 5 μm). BMDMs were incubated with 10 μg/mL of MSP and Eu@MSP for 4 h. The cortical F-actin and nucleus were stained using Rhodamine Texas Red and DAPI, respectively. (**b**) MSP and Eu@MSP uptake observation of BMDMs by live cell holotomography. BMDMs were incubated with 10 μg/mL of MSP and Eu@MSP for 4 h under live cell holotomography. White triangles indicate MSP particles. Red triangles indicated Eu@MSP.

**Figure 8 nanomaterials-11-00343-f008:**
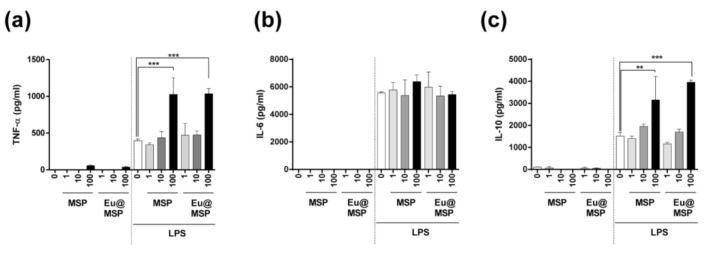
Effects of MSP and Eu@MSP on the production of cytokines of bone marrow-derived macrophage with or without lipopolysaccharide stimulation. The production of TNF-α (**a**), IL-6 (**b**), and IL-10 (**c**) in the supernatant was measured by ELISA. BMDMs were treated with or without LPS (100 ng/mL) and MSP and Eu@MSP for 24 h at the indicated concentration. All data represented the mean ± standard deviation, n = 4 (** < 0.05 and *** < 0.001).

## Data Availability

The data presented in this study are available on request from the corresponding authors.

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
