# Peer review of "Real-Time Tracking of Highly Luminescent Mesoporous Silica Particles Modified with Europium β-Diketone Chelates in Living Cells"

_nanomaterials, 2021, doi:10.3390/nano11020343_

Round 1

Reviewer 1 Report

This paper deals with tracking of luminescent silica particles with Eu-beta dicetonate chelates in living cells. The investigation contains interesting data about preparation, characterisation and optical (bio) behavior of hybrid nano(micro) composites containing Ln complexes. The paper can be published after MINOR REVISIONS. A short list of remarks (R) and questions (Q) is listed bellow:

R1 - Usually Eu - emission spectra are presented in the range 550 - 720 nm, to

display all Eu - transitions

R2 - Excitation spectra, preparation route /in situ/ and thermal properties displayed here are similar to that of other more simple Eu-complexes; and SiO2:Eu-complexes not cited here: JLumin 183 (2017) 108; Opt Mater 30 (2011) 1715; Russ. J. Coord Chem. 34 (2002) 1574

Q1 - what is the origin of the Eu -sharp excitation spectrum? It could be absorption, without energy transfer, for example. Please, add a short explanation.

Author Response

R1 - Usually Eu - emission spectra are presented in the range 550 - 720 nm, to display all Eu – transitions

Ans: Thanks for the comments. I fully agree on your comments. In the case of Eu(TTA)3(P(Oct)3)3, we can plot the emission spectra of 550 - 720 nm. However, unexpected sharp and intense peak is appeared around 700 nm when the Eu(III) complexes are incorporated with spherical particles, such as polystyrene or silica. We try to explain these phenomena, but we could not find corresponding references. So, we just draw the graph similar to other references.   

R2 - Excitation spectra, preparation route /in situ/ and thermal properties displayed here are similar to that of other more simple Eu-complexes; and SiO2:Eu-complexes not cited here: JLumin 183 (2017) 108; Opt Mater 30 (2011) 1715; Russ. J. Coord Chem. 34 (2002) 1574

Ans: Thanks for the comments. Two articles, JLumin 183 (2017) 108; Opt Mater 30 (2011) 1715, were cited in manuscript, but I could not find the article, Russ. J. Coord Chem. 34 (2002) 1574.

Q1 - what is the origin of the Eu -sharp excitation spectrum? It could be absorption, without energy transfer, for example. Please, add a short explanation.

Ans: Thanks for the comments. It is correct that the sharp peak in excitation spectrum is due to the absorption of spherical nanoparticles. This short information was added in sentence. The following sentence was added. “The sharp excitation peaks appeared at 308 nm is coming from the absorption of spherical silica nanoparticles not from the energy transfer from europium complexes.”

Reviewer 2 Report

The manuscript proposed by Kim et al. focuses on the development of the synthesis and the use of high quantum yield europium complexes modified mesoporous silica particles to be used as tracker in living cells. 

The manuscript is explaining how the team synthesized mesoporous silica particles (MSP) in which was incorporated the β-diketone europium complexes inside of their nanocavities in order to allow real time imaging in living cells such as macrophage or mouse bone marrow cells. 

The manuscript describes very avvuratley the synthesis process, the interaction between the MSP and TMSPMA. the adjustment of luminescence detection in terme of excitatioinnand emission are detailed as well as the the effect of Eu@MSP in comparison with MSP on the cytotoxicity of various kinds of eukaryotic Cells.

Luminescence imaging and holotomography show the uptake of the Eu@MSP within the cells.

In overall, the article is clearly written and the data shown validate the methodological process allowing to use Eu@MSP as fast tracks within living cell. I have no major comments on the manuscript except that a stronger conclusion and perspective on the application and developments of such approached could be more detailed.

Author Response

In overall, the article is clearly written and the data shown validate the methodological process allowing to use Eu@MSP as fast tracks within living cell. I have no major comments on the manuscript except that a stronger conclusion and perspective on the application and developments of such approached could be more detailed.

Ans: Thanks a lot for the comments. Based on referee’s comment, following description was added at the end of the discussion part. “The Eu@MSP can be applicable if the surface of the particles were modified with bio-ligands, i.e. antibodies, aptamers, proteins and carbohydrates etc., in the field of bioimaging probes both in-vivo and in-vitro tracking agents. Especially, we are expected that Eu@MSP shield the antigens coming from enzymatic degradation together with fast denaturation, resulting in an extended discharge of antigen for long-term humoral response, or target oriented controlled release for specific cell immunity. In addition, Eu@MSP could enhance the antigen uptake by antigen-presenting cells (APC), promote a stronger immune response resulting in improving therapeutic effects.”

Reviewer 3 Report

Title: Real-time tracking of highly luminescent mesoporous silica 2 particles modified with europium β-diketone chelates in living 3 cells

Authors: Jong-Seok Kim, Sung Ki Lee, Hansol Doh, Do Kyung Kim

In this manuscript, the authors present the development of europium complexes modified with mesoporous silica particles (MSP) with potential applications as imaging probes for both in-vitro diagnostic and in-vivo cellular tracking agents. In this paper the authors present the results of phisico-chemical investigation (by scanning electron microscopy, transmission electron microscopy, FTIR-ATR spectroscopy, photoluminescence, DLS measurements and Thermal gravimetric analysis (TGA) and differential 123 scanning calorimetry (DSC) analysis) conducted on the obtained samples. Moreover, information regarding the cytotoxicity of MSP and Eu@MSP to A549, HeLa, mouse bone marrow cells and mouse bone marrow-derived 342 macrophage are presented in the paper. Also, the effects of MSP and Eu@MSP on the production of cytokines of bone marrow-derived macrophage with or 365 without lipopolysaccharide stimulation are reported.

In my opinion, this manuscript could be considered for publication, after the following minor changes:

  1. In the FTIR spectra (figure 3) the peaks must be indexed.

  2. Furthermore, an EDS spectra of the samples should be added in the manuscript.

  3. Also, in the figure 5 the maxima should be indexed.

  4. Also, when you present and discuss the results please see and refer to: https://doi.org/10.1155/2013/284285, https://doi.org/10.1155/2012/942801 and https://doi.org/10.2478/s11532-014-0554-y .

Author Response

In my opinion, this manuscript could be considered for publication, after the following minor changes:

  1. In the FTIR spectra (figure 3) the peaks must be indexed.

Ans: Thanks for the comments. Following explanation was added in sentence. “The peaks at 1410 and 1352 cm-1 are assigned to ν(C=C, C=S thienyl heterocycle). CF3 in TTA moiety vibration peaks are appeared at 1292 cm-1 (ν(CF3)) and 723 cm-1 (δ(CF3)). The vibration peaks of keto-enol tautomerization of β-diketone ligand can be assigned to 1535 cm-1 (ν(C=O)) and 1516 cm-1 (ν(C=C)). The peaks at 1064 cm-1 (δ(O-CH3)) , 861 cm-1 (δ(CH3)) and 797 cm-1 (δ(CH, thienyl)). “

  1. Furthermore, an EDS spectra of the samples should be added in the manuscript.

Ans: Thanks for the comments. However, we don’t have EDX in our institute. It takes some time in current situation. Editor requests us to submit revision within 3 days, so we could not handle this on time. We are very sorry for the situation.

  1. Also, in the figure 5 the maxima should be indexed.

Ans: Thanks for the comments. The peaks are indexed.

  1. Also, when you present and discuss the results please see and refer to: https://doi.org/10.1155/2013/284285, https://doi.org/10.1155/2012/942801 and https://doi.org/10.2478/s11532-014-0554-y .

Ans: The articles were cited in introduction part.